# Social support, social network and salt-reduction behaviours in children: a substudy of the School-EduSalt trial

Yuan Ma,[1,2] Xiangxian Feng,[3] Jun Ma,[4] Feng J He,[5] Haijun Wang,[4] Jing Zhang,[6] Wuxiang Xie,[7] Tao Wu,[6] Yunjian Yin,[8] Jianhui Yuan,[3] Graham A MacGregor,[5] Yangfeng Wu[1,6,7]

**Correspondence to**
Dr Yangfeng Wu;
wuyf@bjmu.edu.cn

## ABSTRACT

**Objectives** Healthy behaviour changes, such as reducing salt intake, are important to prevent lifestyle-related diseases. Social environment is a major challenge to achieve such behaviours, but the explicit mechanisms remain largely unknown. We investigated whether social networks of children were associated with their behaviours to reduce salt intake.

**Design** An ancillary study of a school-based cluster randomised controlled trial to reduce salt intake in children and their families (School-EduSalt), in which salt intake of children was significantly reduced by 25%.

**Setting** 14 primary schools in urban Changzhi, northern China.

**Participants** 603 children aged 10–12 years in the intervention arm.

**Primary and secondary outcome measures** We developed a score assessing salt-reduction behaviours (SRB score) of children based on self-administered questionnaires. The SRB score was validated by the changes in salt intake measured by 24-hour urine collection in a random sample of 135 children. A 1-unit increase in SRB score was associated with a 0.31 g/day greater reduction in salt intake during the trial (95% CI 0.06 to 0.57, p=0.016).

**Results** Children from families with more family members not supporting salt reduction had significantly lower SRB scores (p<0.0001). Children from a class with a smaller size and from a class with more friendship connections, as well as children having more friends within the class all showed higher SRB scores (all p<0.05). Children whose school teachers attended the intervention programme more frequently also had higher SRB scores (p=0.043).

**Conclusion** Social networks were associated with the behaviours to reduce salt intake in children. Future salt-reduction programmes may benefit from strategies that actively engage families and teachers, and strategies that enhance interconnectivity among peers.

**Trial registration number** NCT01821144; post-results.

## Strengths and limitations of this study

► Leveraging a randomised controlled trial that demonstrated a successful intervention strategy to reduce salt intake in children and their families, we identified the social network factors associated with behaviours to reduce salt intake in children.
► The validity of the primary outcome was validated by the change in sodium intake measured by 24-hour urine collections.
► Advanced social network analysis techniques in addition to the traditional methods were used to carefully examine the role of peer influence in salt reduction.
► The sensitivity analysis, using change in salt intake measured by 24-hour urinary sodium as the outcome, showed consistent results, suggesting the robustness of the major findings.
► Given the exploratory nature of the study, data collected after the end of the main trial were subject to recall bias. Temporal order of the association warrants future study.

## BACKGROUND

Cardiovascular disease (CVD) is a major cause of death globally despite substantial improvement in CVD treatment.[1] Dietary risk factors are the largest contributors to global disease burden, with high sodium intake among the top ones.[1] Compelling evidence also highlights the crucial role of dietary and lifestyle-related behaviour changes in curbing global CVD pandemic,[2–4] but factors driving effective behavioural interventions are not clear.[5–7]

Evidence from observational studies suggests that social networks (ie, social ties connected to the individual) may affect individuals' health behaviours such as obesity, smoking, fruit and vegetable consumption, partly through social support and social influence.[8–10] Data from interventional studies also demonstrate that intervention on tobacco control incorporating the role of network structure and social support is more effective than usual interventions.[11–14] Emerging evidence in children also emphasises the role of peer influence in children's eating behaviours.[15 16] However, evidence examining

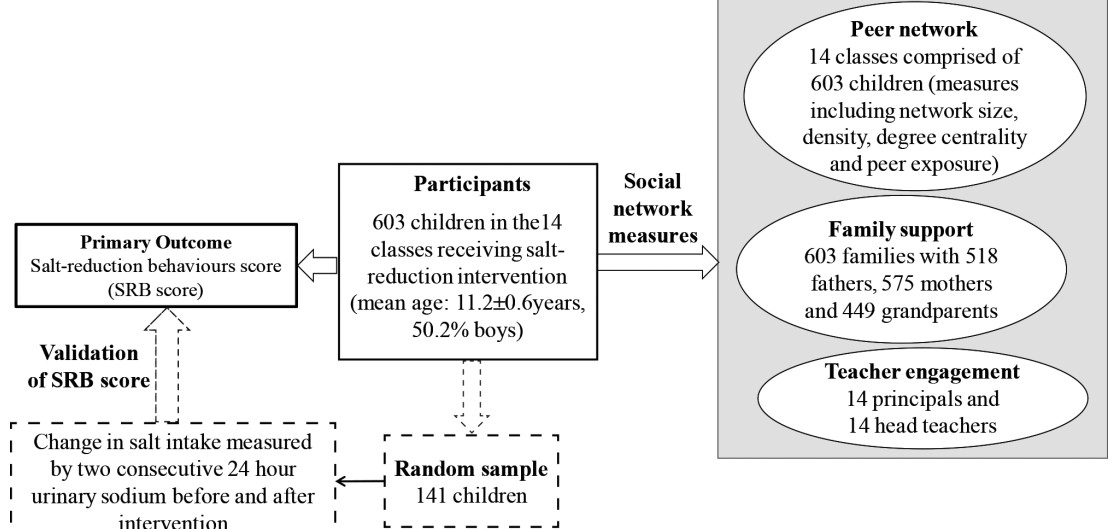

**Figure 1** School-EduSalt network study design.

role of social networks in behaviours to reduce salt intake is lacking.

Salt intake is high in children in most countries,[17–20] and it is shown that greater long-term health benefits can be achieved through a lower salt diet initiated early from childhood.[21 22] School-EduSalt trial is one of the few randomised trials worldwide that demonstrated a successful reduction of 25% in salt intake among children and families through a school-based education programme.[20] We hypothesised that social networks could play an important role in affecting effective behaviour changes of children to reduce salt intake. Therefore, the current study determined whether the three domains of children's social networks, including peers, families and teachers, were associated with the magnitude of salt reduction among children in the intervention group.

## METHODS
### The School-EduSalt trial
School-EduSalt (Based Education Programme to Reduce Salt) was a cluster-randomised controlled trial carried out in 28 primary schools in an urban area of northern China. One class in grade 5 was selected from each school. The 28 classes were randomly allocated (1:1) to either intervention or control group. School-based salt-reduction education programme was delivered to all the children and their families in the 14 classes of the intervention group, while no specific intervention was delivered to the control group. The duration of the intervention was a school semester (≈3.5 months). Around 10 children were randomly selected from each class and 2 adult family members of each of these children were invited to collect two-consecutive 24-hour urine at both baseline and the end of trial. A detailed report of the study design and the main results can be found elsewhere.[20 23] Only brief methods related to this study were described here.

### The children social networks study
The subjects of this study were all the schoolchildren aged 11.2±0.6 (SD) years in the intervention group. The primary outcome was the score on salt-reduction behaviours (hereinafter referred to as SRB score), validated by the reduction in salt intake measured by 24-hour urinary sodium excretion before and after the intervention. Primary exposure measurements were factors related to the three major domains of the networks of children, including family support, peer network and teacher engagement. Children's peer networks were sociocentric networks with each of the 14 classes being a complete network, and all 603 children receiving intervention were recruited. For family and teacher network, only family support and teacher engagement were assessed due to data availability.

### Participant and public involvement
The school principals and head teachers of the included schools were involved in the study design stage for the development of intervention activities and for the feasibility assessment of the main trial, from which the present study was further developed. No participants were involved in setting the research question or the outcome measures for the present study, nor were they asked to advise on the interpretation or the writing up of this study.

### Data collection
Data used in this study were collected throughout the main trial, together with an additional post-trial survey specifically designed for this study (figure 1). The post-trial survey was carried out among all the children in the intervention classes 9 months after the end of the intervention. In this survey, children were asked to fill out a self-administered questionnaire at school time for data on children's peer network, their recall of their family members' support for the School-EduSalt intervention programme during the trial period, and behaviours

related to salt reduction. Almost all children (603 out of 604) completed the survey. In the main trial, two-consecutive 24-hour urine for each child, both before and after the intervention, was collected among a random sample of 141 children to estimate the change in salt intake.[20 23] More details are given below.

## Outcome assessment

The SRB score was derived from three questions on behaviours related to salt reduction that were collected in the post-trial survey. The three questions were the frequency of eating high-salt snacks (≥3 days/week, 1–3 days/week, almost never), the frequency of eating high-salt pickles (≥3 days/week, 1–3 days/week, almost never) and whether their families are taking actions to reduce salt intake (never, used to, now). Scores ranging from 1 to 3 were assigned to the answer of each question in order. The SRB score was calculated by summing up the standardised score (ie, the z-score, to make the weight of each question comparable) from each question.

To validate SRB score, we assessed the association of SRB score with the reduction in salt intake measured by 24-hour urinary sodium excretion before and after the intervention among a random sample of children who made 24-hour urine collection as described above. Mixed linear models were used with school and class as random effects, with additional adjustment for age, gender and baseline body mass index. The 24-hour urine collections deemed incomplete were excluded from the analyses.[20] The number of children with both valid measurements of social network and completed 24-hour urine collection was 135.

## Assessment of network measures

Family support for the salt-reduction intervention from each family member was assessed by asking all children in the post-trial survey (ie, supporting, opposing, neither supporting nor opposing). Children's peer network was defined as friends within the same class. Each child (ie, the nominator) was asked to nominate his/her friends (ie, the nominees) from the same class who met at least two of the following conditions: (1) play together almost every day during school time, (2) often play or do homework together after school time (>3 days/week), (3) often share snacks and toys, (4) often share extracurricular books and (5) often go to school or go home after school together. In order to take the strength of relation ties into account, we examined friendship ties weighted by the number of above conditions that were met. Peer network measures also included the size and density of the network, as well as degree centrality and peer exposure of each child. Specifically, network size is the number of children in each class. Network density is the total number of friendship connections (ties) divided by the total possible number of ties within the same class, which reflects the overall interconnectedness of the children in the same class.[24] Degree centrality is the total number of friends the child had, and it reflects the degree to which the child is connected to the network.[24] We used normalised degree centrality that was divided by the size of class in order to

make it comparable across classes. Both density and degree centrality reflect the connectivity of a network, the former at network level and the latter at individual level. Peer exposure (peers' salt-reduction behaviours) of a child was calculated by summing and averaging the SRB scores of his/her friends. Teacher engagement was recorded during the intervention and was defined as full engagement (both the principal and headteacher attended all activities they had been invited to), partial engagement (attending some of the activities) and no engagement (attending none of the activities).

## Data analysis

The association of family support with SRB score was assessed using mixed linear models, which accounted for the clustering effect of children nested within the same class, with the adjustment for age and gender.

For measures on peer networks, their association with SRB score was first assessed using mixed linear models, with the adjustment for age and gender. Peer networks of all children and their SRB scores were then visualised using network mapping techniques. Additionally, to examine whether SRB scores of the children were influenced by the SRB level of their friends (ie, peer influence), analysis was performed from individual-focused and relation-focused facet, respectively. We first built traditional (individual-focused) mixed models on children's SRB score, with children's peer exposure, age and gender as covariates. This traditional method using mixed models considered the dependence of observations, but it may not be the optimal method to address the dependence due to the overlap of peer exposures. To address this issue and make inference from network structures, we then used exponential random graph models (ERGM).[25] The ERGM of social network structure was fit for each peer network based on the difference in SRB scores between each pair of friend ties. This model explained whether the probability of the existence of friend ties between two children was associated with the similarity in SRB scores.[26] We applied the same model to each of the 14 classes and then used meta-analysis to pool the results across classes.[27 28] Fixed-effect model would be used in the absence of significant heterogeneity as assessed by $I^2$ statistics, otherwise random-effect model would be used.

Teacher engagement was examined using methods consistent with those for family support. To additionally explore which network (family, peer and teacher) would be potentially more important in influencing the salt-reduction behaviours of children, as a secondary analysis, we examined the role of children's social networks simultaneously by including a representative factor for each network in the mixed linear model.

## Sensitivity analysis

To test the robustness of primary findings, we performed a sensitivity analysis in a random sample of children (n=135) with completed 24-hour urine collection at both baseline and the end of the trial. Specifically, change in salt intake (latter-former), assessed by 24-hour urine

sodium excretion, was used as the outcome. The association of three network components (ie, peer, family and teacher network components) with change in salt intake of children were investigated using mixed models with adjustment for age, gender and baseline body mass index.

Traditional data analysis was performed in SAS V.9.4. Network properties were calculated in Ucinet V.6.0[29] while networks were visualised in NetDraw V.2.155.[24 30] ERGM models were fit using the *statnet* package in R environment.[22 25]

## RESULTS

### Participant characteristics

The average age of the 603 children (boys: 50.2%) was 11.2 (mean) ±0.6 (SD) years old. Almost half (46%) gained full support from their families for the School-EduSalt salt-reduction programme. The proportion of boys whose fathers did not support the programme was higher compared with girls (p=0.0035). Around 50% of the children had teachers fully engaged in the intervention programme (table 1). The average size of the classes was 43 (range: 13–77). Salt-reduction behaviours of children are described in table 2.

### Validation of the SRB scores

In the random sample of 135 children with complete 24-hour urine collection both before and after intervention, the validation analysis showed that every one-point increase in SRB score was associated with a 0.31 g/day (95% CI 0.06 to 0.57, p=0.016) greater reduction in salt intake after adjusting for age, gender and baseline body mass index. As further illustrated in figure 2, the reduction in salt intake increased gradually from the lowest to the highest quartile of SRB score, presenting a significant linear trend (p trend=0.036), with a greater SRB score indicating a larger (favourable) reduction in salt intake.

### Measures of social network and SRB scores

From the perspective of family networks, the SRB score of children was less with more family members not supporting salt reduction (p for trend <0.0001). The SRB score of children whose father did not support salt reduction was significantly lower compared with the children whose father did (p<0.001). Similar association was observed for the support from mother. The magnitudes of the associations appeared smaller for the support from grandparents (table 3).

From the perspective of peer networks, smaller size, higher density and higher normalised degree centrality were significantly associated with a larger SRB score after adjusting for children's age and gender (table 3). Children's peer networks were further visualised in figure 3, showing that children of greater scores (ie, greater nodes) appeared to have denser local networks. Furthermore, traditional analysis shows that every 1-unit increase in the SRB score of the peers was associated with a slight and non-significant increase of 0.07 (95% CI −0.07 to 0.20,

| Table 1 | Participant characteristics |
|---|---|
| **Characteristics** | **All (n=603)** |
| Age (years) | 11.2±0.6 |
| Boy, n (%) | 303 (50.2) |
| Family support, n (%) | |
| Family members not supporting salt reduction (number) | |
| ≥3 | 72 (12.0) |
| 2 | 91 (15.1) |
| 1 | 160 (26.6) |
| 0 | 279 (46.3) |
| Family members not supporting salt reduction (roles)* | |
| None | 279 (46.3) |
| Father | 154 (25.6) |
| Mother | 79 (13.1) |
| Grandmother | 90 (14.9) |
| Grandfather | 74 (12.3) |
| Peer network measures, n (%) | |
| Network size (No of children within the class)† | 43±16 |
| Network density (in %)† | 11±7 |
| No of friends | 4 (2–5) |
| Normalised degree centrality (in %)‡ | 24 (12–43) |
| Peer exposure (score on salt-reduction behaviours) | 0.04 (−0.70 to 0.80) |
| Teacher engagement, n (%) | |
| Full | 311 (51.6) |
| Partial | 292 (48.4) |

Data are shown in the format of mean±SD, median (P25–P75) and n (%).
*The overall proportion exceeded 100% as there were cases in which two or more family members did not support salt reduction, that is, these categories were not mutually exclusive.
†Measures at cluster (class) level. Density of network is the total number of friendship connections (ties) divided by the total possible number of friendship ties within the same class.
‡Normalised degree centrality is the number of friends within the class divided by the size of class.

p=0.21) in the SRB score of children (table 3). Consistently, according to meta-analysis of the 14 ERGM built for each of the 14 networks, similarity in SRB score among friends was not found (OR=1.02, 95% CI 0.93 to 1.11, p=0.68, online supplementary figure S1).

In terms of teacher engagement, the SRB scores of the 311 children, from schools where both the principal and head teacher fully participated in the intervention, were significantly higher, compared with their counterparts from the other schools (difference in SRB score: 0.33; 95% CI 0.01 to 0.64, p=0.043, table 3).

When network factors from the above three domains were examined in the same model, the role of family support remained statistically significant (p<0.0001), The association with the degree of being connected to

| Table 2 Description of children's salt-reduction behaviours | |
|---|---|
| Questions related to salt reduction (score assigned for each answer) | All (n=603) |
| Q1. Frequency of eating pickles, N (%) | |
| Almost never (=3) | 446 (74.0) |
| 1~3 days/week (=2) | 147 (24.3) |
| ≥3 days/week (=1) | 10 (1.7) |
| Q2. Frequency of eating high-salt snacks, N (%) | |
| Almost never (=3) | 166 (27.6) |
| 1~3 days/week (=2) | 358 (59.3) |
| ≥3 days/week (=1) | 78 (13.0) |
| Q3. Family's action on salt reduction, N (%) | |
| Have been reducing salt (=3) | 401 (66.6) |
| Used to (=2) | 172 (28.6) |
| Never (=1) | 29 (4.8) |
| Sum of Q1–Q3 scores, N (%) | |
| 7–9 | 95 (15.8) |
| 4–6 | 482 (80.2) |
| 0–3 | 24 (4.0) |
| SRB score (the sum of standardised scores) (mean±SD) | 0±1.9 |

| Table 3 Association of SRB score with social network measures among all 603 children | | |
|---|---|---|
| Network measures | Regression coefficient (95% CI)* | P value |
| Family support | | |
| Family member not supporting salt reduction (number)† | | |
| ≥3 | −1.1 (−1.6 to −0.6) | <0.0001 |
| 2 | −1.0 (−1.4 to −0.5) | <0.0001 |
| 1 | −0.6 (−0.9 to −0.2) | 0.003 |
| 0 | Reference | – |
| Family member not supporting salt reduction (roles) | | |
| Father | −1.3 (−1.6 to −0.9) | <0.0001 |
| Mother | −1.2 (−1.8 to −0.6) | 0.002 |
| Grandmother | −0.7 (−1.4 to −0.1) | 0.024 |
| Grandfather | −0.8 (−1.5 to 0.02) | 0.06 |
| None | Reference | – |
| Peer network measures | | |
| Network size (No of children within the class) | −0.01 (−0.02 to −0.002) | 0.026 |
| Network density (in %) | 0.94 (0.06 to 1.82) | 0.036 |
| Normalised degree centrality (in %) | 0.5 (0.01 to 0.99) | 0.044 |
| Peer exposure (score on salt-reduction behaviours) | 0.07 (−0.07 to 0.20) | 0.205 |
| Teachers' engagement | | |
| Partial | −0.33 (−0.64 to −0.01) | 0.043 |
| Full | Reference | - |
| Joint association | | |
| Family support (partial versus full)‡ | −0.82 (−1.13 to −0.52) | <0.0001 |
| Peer network (normalised degree centrality) | 0.37 (−0.11 to 0.85) | 0.13 |
| Teacher engagement (partial versus full) | −0.25 (−0.56 to 0.07) | 0.20 |

*Linear regression coefficients for corresponding network measures in the mixed models on SRB score.
†P for trend <0.0001.
‡Full support was defined as all family members supporting salt reduction. Partial support was defined as at least one family member not supporting salt reduction.

peers in the class, as measured by the normalised degree centrality, became borderline significant (p=0.13). Findings were similar for size and density of peer networks (data not shown). The association of SRB score with teacher engagement was also attenuated (table 3).

In the final sensitivity analysis replacing SRB score with the change in salt intake measured by 24-hour urinary sodium excretion as the outcome, results were consistent with primary findings. Specifically, with more family members not supporting salt reduction, the reduction in salt intake of children tended to be less (p for trend=0.01). Children whose father did not support salt reduction also had an average of 1.39 g (95% CI 0.38 to 2.41) less reduction in daily salt intake compared with the children with support from all family member (p<0.001). The

role of family support remained statistically significant after further accounting for factors of peer and teacher networks. The association of change in salt intake with peer and teacher network factors were not statistically significant, with point estimates showing essentially similar patterns (online supplementary table S1).

## DISCUSSION
Our study is the first to explore the role of social networks in salt-reduction behaviours. We found that salt-reduction behaviours in children were related to their family support, peer networks and teacher engagements from multiple aspects. These findings offer new insights into

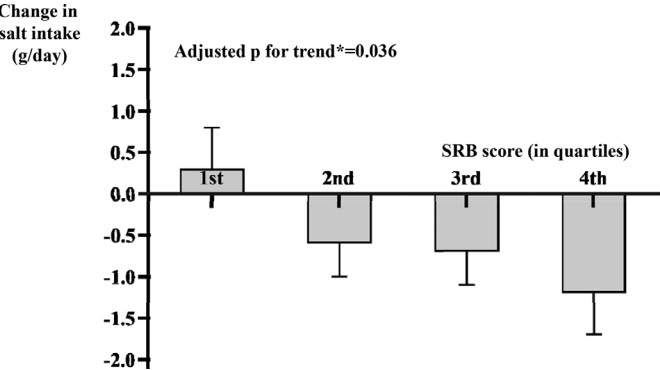

**Figure 2** Change in salt intake measured by 24-hour urinary sodium across the quantiles of SRB score in children.

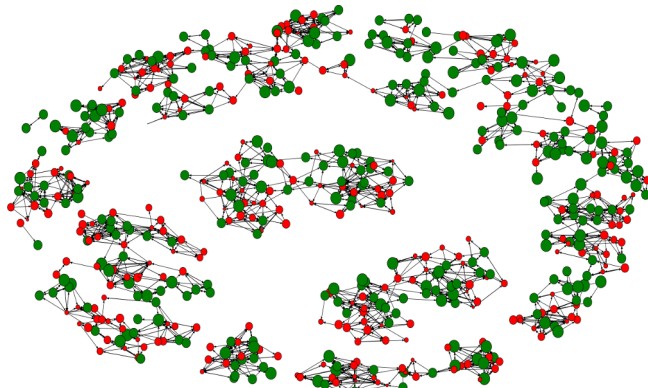

**Figure 3** Visualisation of peer networks in the 14 intervention schools.

the potential mechanisms underlying successful salt-reduction interventions in the School EduSalt trial. In this trial, interventions were implemented at the class level, with activities strategically designed to engage families, peers, teachers and school administrators. This study provides new evidence on how to design effective behaviour change programmes to reduce salt intake from a social network perspective.

Our findings on the role of family networks in salt-reduction interventions were consistent with the main body of evidence that demonstrates the role of family support in shaping smoking, physical activity and dietary behaviours.[31–33] Our results also suggest that family network, especially family support, could play a more important role in shaping the behaviours of children to reduce salt intake, compared with peer and teacher networks. One explanation could be that children recruited in this study usually had meals at home and therefore their family members had more opportunities to exert a significant influence on their eating behaviours. For peer networks, both density and degree centrality, which reflect better interconnectivity of network members, were associated with favourable behaviours to reduce salt intake. It is possible that the networks that are better interconnected could provide children with more opportunities to gain peer support[34] and to influence their peers, which would in turn lead to better salt-reduction behaviours. Likewise, children with more friends within the same class may have a better sense of belonging to the class and were thus more likely to get involved in salt-reduction activities in the class. Similar inverse association between obesity and density of friend networks was also reported in an observational study.[35] By contrast, peer influence and homophily phenomena, reported for obesity[8] and smoking,[9] were not observed for salt reduction in this study. One possibility could be that children with obesity and smoking behaviours are likely to play together but children with similar salt intakes may not necessarily do so. It indicates that, unlike smoking and obesity, salt reduction behaviours of a child may not influence its peer friendship, or vice versa. Furthermore, as first reported in this study, the engagement of

school teachers in the intervention activities may also significantly influence the salt-reduction behaviours of children. Their role might be directly exerted through strengthening the children's attention to the intervention activities or reinforcing their beliefs in the salt-reduction messages delivered during the health education classes. It could also be mediated through their indirect influences on parents so that parents were more actively engaged in the intervention activities.

The role of social networks in shaping attitudes and behaviours related to non-communicable chronic diseases has been increasingly recognised.[36] Emerging intervention studies incorporating network components have achieved desired behaviour changes.[10–13] For example, it was reported that observing videos of heroic peers eating fruit and vegetables increased fruit consumption in children.[10] It was also reported that intervention for smoking cessation delivered by peers was more effective compared with usual care.[12] Furthermore, modifying individual's network structures may enhance the social diffusion of desired behaviours.[37] One study found that assigning intervention groups based on naturally formed network, compared with random assignment, enhanced the effect of tobacco control among children.[11] Similarly, a nutritional intervention that targeting nominated friends of individuals was also demonstrated to increase the adoption of a nutritional intervention.[38] Nevertheless, given the difficulty in implementing effective dietary and lifestyle interventions, the evidence on incorporating individuals' social networks into behaviour change interventions is far from sufficient. This is particularly true considering that non-communicable diseases are rooted in unhealthy lifestyle behaviours. Our study, together with existing evidence,[13] suggests the great potential to promote desired behaviour changes and to effectively prevent non-communicable diseases through assessing the social network context and engaging appropriate network interventions.

Our study has several limitations. Our data on peer networks and family support were collected after the end of the trial, which limited our ability to draw causal conclusions on the association of peer network factors with salt-reduction behaviours. Recall bias is also possible and it is more likely that individuals misreported less favourable support as desired support rather than the other way around, possibly resulting in underestimates of the relationships. Additionally, for family and teacher networks, we only examined family support and teacher engagement due to limited data availability. Finally, although peer exposure was not associated with individuals' behaviours to reduce salt intake in the current study, peer networks beyond the scope of the current study may play a role, such as friends beyond the class and from social media. Our study also has several strengths. This is the first study to explore the role of social network factors in salt-reduction behaviours. This study is based on a well-designed randomised controlled trial that successfully reduced salt intake in children and their families

in the intervention arm by about one-fourth in a school semester. Moreover, the primary outcome, that is, SRB score, was validated by sodium intake assessed by 24-hour urine collection—the current gold method for measuring salt intake—in a random sample of children.[39] Though an ideal solution would be assessing behaviours using 24-hour urine collection for children, it is not feasible to implement this on a large scale as it is costly, resource consuming and challenging to ensure the completeness of 24-hour urine collection.

## CONCLUSION

Our study demonstrates that social networks may play an important role in shaping the behaviours to reduce salt intake in children. It provides new evidence for the benefit of integrating social network components into behaviour change interventions. It has a significant implication for the development of salt-reduction strategies. In a broader context, the study also sheds light on the great potential of incorporating social networks in other diet and lifestyle interventions for the prevention and control of lifestyle-related diseases.

**Author affiliations**
[1]Peking University Health Science Centre, Department of Epidemiology and Biostatistics, Beijing, China
[2]Department of Epidemiology, Harvard University T.H. Chan School of Public Health, Boston, Massachusetts, USA
[3]Department of Preventive Medicine, Changzhi Medical College, Changzhi, China
[4]Institute of Child and Adolescent Health, Peking University Health Science Center, Beijing, China
[5]Wolfson Institute of Preventive Medicine, Queen Mary University of London, London, UK
[6]The George Institute for Global Health at Peking University Health Science Center, Beijing, China
[7]Peking University Clinical Research Institute, Beijing, China
[8]Department of Probability and Statistics, Peking University School of Mathematical Sciences, Beijing, China

**Acknowledgements** We thank the children and their families, the head teachers and school principals who participated in the study. We also thank all the team members from Changzhi Medical College, particularly Jianhui Yuan, Yanbo Han, Zhifang Li, Jianbin Zhang, Peifen Duan, Cailing Wei, Yanli Zhai, Ziqun Zhuo, Meng Xiao and Yunbin Zhang for their help with the data collection.

**Contributors** FJH, YW, XF, JM, HW and GAM designed the main study (School-EduSalt trial). YW and YM designed this substudy and developed the analysis plan. XF, YM, JY and JZ contributed to data collection. YM, YY, TW and WX performed statistical analyses. YM, YW, FJH, WX, YY, TW, HW and GAM contributed to the interpretation of data. YM and YW took responsibility for the integrity of the data and the accuracy of the data analysis. YM wrote the first draft of the manuscript. YW, FJH, XF, JM, HW, JZ, WX, TW, YY, JY and GAM critically reviewed the manuscript. All authors approved the final version. YM and YW are guarantors.

**Funding** The study was funded by the UK Medical Research Council (MR/J015903/1).

**Disclaimer** The funder of the study had no role in the design of the study; the collection, analysis and interpretation of the data; the writing of the manuscript; and the decision to submit the article for publication.

**Competing interests** None declared.

**Patient consent for publication** Not required.

**Ethics approval** This study was approved by Peking University Health Science Centre Institutional Review Board (IRB00001052-12072) and Queen Mary University of London Research Ethics Committee (QMREC2012/81).

**Provenance and peer review** Not commissioned; externally peer reviewed.

**Data sharing statement** The datasets used during the current study are available from the corresponding author on reasonable request.

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
