## [Reviewer comments · BMJ Open]

ARTICLE DETAILS

TITLE (PROVISIONAL)	Social support, social network and salt-reduction behaviours in children: a sub-study of the School-EduSalt trial
AUTHORS	Ma, Yuan; Feng, Xiangxian; Ma, Jun; He, Feng; Wang, Haijun; Zhang, Jing; Xie, Wuxiang; Wu, Tao; Yin, Yunjian; Yuan, Jianhui; MacGregor, Graham; Wu, Yangfeng

VERSION 1 - REVIEW

REVIEWER	Tatsuo Shimosawa International University of Health and Welfare Japan
REVIEW RETURNED	27-Jan-2019

GENERAL COMMENTS	The authors analyzed School-EduSalt trial to find social environment impact on salt reduction in children and address important issue for population approach to reduce salt intake. The study is well organized and data are properly analyzed except for one minor concern I have. In Method section, the authors stated that both children and parents urine were analyzed for sodium excretion but I could not find data of urinary sodium excretion changes in parents. Does it correlate with children's urinary sodium excretion? What was the impact of father's involvement?
---

REVIEWER	Carley Grimes Deakin University, Australia
REVIEW RETURNED	14-Feb-2019

GENERAL COMMENTS	This is an interesting paper which makes use of existing data collected within a randomised trial targeting dietary salt reduction among Chinese families. The data presented are novel, indicating that select social support structures are important factors to help children change behaviours to lower dietary salt intake. This information can be used in the development of future salt reduction education and behavioural programs. The paper is well written and information clearly presented. I have some minor comments to address below. • Can it please be clarified in the methods when the questions for the SRB score were administered to children. Was this done just once, at the end of the study? i.e. there was no pre score and therefore change in this score? (Unlike salt intake for which there
--

	is a change score). Were the Qs included for SRB score part of the original planned trial and data collection?  • Line 6, data analysis section: additional adjustment of BMI too across quartile SRB score • Line 53: Is patient ? a typo, perhaps participant • Table 1& 3: why is it not possible to separate out mother from grandparents i.e. the 'others'; fathers was collected separately and information can then be provided about the importance or not of support from this family member. Is there a reason why mothers was not specifically examined? Can this be stated in the method. Particularly as this is also a discussion point at line 41, page 12
--	---

REVIEWER	Walter Lehmacher Uni Köln
REVIEW RETURNED	26-Feb-2019

GENERAL COMMENTS	The reviewer completed the checklist but made no further comments.
--

REVIEWER	Gabriel Escarela Universidad Autónoma Metropolitana - Iztapalapa
REVIEW RETURNED	06-Mar-2019

GENERAL COMMENTS	The methods are correctly used. I recommend to give the references of the techniques used and not only the software employed.
---

REVIEWER	Yipu Shi Public Health Agency of Canada Canada
REVIEW RETURNED	25-Mar-2019

GENERAL COMMENTS	Comments to the statistical procedures:  1. Well designed and methodological vigour school based intervention study. 2. Mixed linear models were used to appropriately account for the clustering effect of children in the same class. It's a methodological strength however, it seems that the association of SRB with each aspect of the children's social networks was assessed individually, then the joint effect of these networks on SRB was assessed in a new model by choosing one most influencing variable from each network (page 8 line 46), can't this be done in one mixed model so the independent effect of all variables of the network are assessed simultaneously? 3. It's not clear how the family support (partial vs. full) was defined in the joint mixed linear model, looks like it's defined differently in the individual family support model (Table 3) 4. Are the children having their 24-hour urine collected before and after the trial used for the validation of the SRB score (n=135) the same as those used for the sensitivity analysis (n=135)? Suggest the reduction in 24-hour urine used in the mixed model in
---

	Supplementary table S1 be treated as positive so that the direction and strengths of the association are comparable with Table 3. 5. The authors used advanced statistical model and social network analysis techniques in addition to the traditional method to carefully examine the role of peer influence in salt reduction, which is another technical strength and contribution to the methodologies in this field.
--	---

VERSION 1 – AUTHOR RESPONSE

Reviewer #1 Tatsuo Shimosawa, International University of Health and Welfare, Japan

The authors analyzed School-EduSalt trial to find social environment impact on salt reduction in children and address important issue for population approach to reduce salt intake.

1.The study is well-organized and data are properly analyzed except for one minor concern I have. In Method section, the authors stated that both children and parents urine were analyzed for sodium excretion but I could not find data of urinary sodium excretion changes in parents. Does it correlate with children's urinary sodium excretion?

Response: We thank the reviewer for raising this relevant point. In the main trial, 24-hour urine collection was made for children and two adult family members of each child at both baseline and the end of the trial. In the current study, we did not provide data on the change in sodium intake of parents because this study aimed to examine the social network factors associated with sodium consumption of children.

To answer the reviewer's question, we have performed the following analyses. The change in salt intake in children from baseline to the end of the trial (post-pre) was -0.7g/d (95% confidence interval: -1.1 to -0.3). The change in salt intake in adults was -2.2 gram/d (95% confidence interval: -2.9 to -1.6). The change in sodium intake of children was closely related to the average change in sodium intake of their family members, and the correlation coefficient was 0.70 (P<0.001). Among the family members recruited, 73% were parents and 27% were grandparents. Sodium intake of children and that of their parents were also significantly correlated, but with smaller magnitude (correlation coefficient r=0.40, P<0.001).

2.What was the impact of father's involvement?

Response: We thank the reviewer for this relevant comment. The father's involvement, assessed by the support for salt reduction, had a significant and positive impact on the sodium reduction of children. As shown in table 3, children whose father did not support salt reduction had a significantly lower SRB score, compared to the children whose father did (Difference in SRB score: -1.3; 95%CI: -1.6 to -0.9, P<0.001), where a lower SRB score indicated a smaller (unfavorable) reduction in salt intake. Consistently, in the sensitivity analysis among the 135 children who had 24-hour urine collection, children whose father did not support salt reduction had a 1.4g/d (95%CI: 0.4 to 2.4) less reduction in salt intake, compared to children whose father supported salt reduction (supplementary table S1).

Reviewer #2 Carley Grimes, Deakin University, Australia

This is an interesting paper which makes use of existing data collected within a randomised trial targeting dietary salt reduction among Chinese families. The data presented are novel, indicating that select social support structures are important factors to help children change behaviours to lower dietary salt intake. This information can be used in the development of future salt reduction education and behavioural programs. The paper is well-written and information clearly presented. I have some minor comments to address below.

1. Can it please be clarified in the methods when the questions for the SRB score were administered to children. Was this done just once, at the end of the study? i.e. there was no pre score and therefore change in this score? (Unlike salt intake for which there is a change score). Were the Qs included for SRB score part of the original planned trial and data collection?

Response: We thank the reviewer for this important comment. The post-trial survey, in which information on SRB score was collected, was conducted nine months after the end of the trial. This information is provided in the method section (Line 6-11 on Page 6). We do not have data on SRB score at baseline because this study was proposed after the main trial was completed, as an ancillary study of the main trial, with the aim to better understand why the intervention worked for some children but failed for others so that future interventions can be better informed. This study was not part of the original planned trial. Clearly, this is not a perfect design, but given the time it took to develop the protocol, to obtain ethical approval, and to coordinate field work, we had strived to ensure a rigorous and timely implementation. During the stage of data analyses, we carefully assessed the validity of the SRB score using the change in salt intake measured by the gold-standard method, i.e. two consecutive 24-hour urinary sodium excretions. We also performed the sensitivity analysis using change in sodium intake as the outcome, which showed consistent findings, suggesting the robustness of the findings. Nevertheless, this is a limitation of our study, and we have acknowledged this in the discussion section (Line 26-28 on Page 13 and line 1-3 on Page 14).

2. Line 6, data analysis section: additional adjustment of BMI too across quartile SRB score

Response: We thank the reviewer for this relevant comment. Our primary analysis was performed among all 603 students within the 14 classes. We didn't adjust for BMI (body mass index) because weight and height were measured only for a subsample of 135 children who collected 24-hour urine. To account for the potential confounding effect of body weight, we performed the following analyses. First, the SRB score was validated by the change in salt intake from baseline to the end of trial measured by 24h urine, with the adjustment of age sex and BMI. The SRB score was significantly associated with the reduction in salt intake ($\beta=0.31$, 95%CI: 0.06-0.57, $P=0.016$), suggesting that a higher SRB score was associated with a larger reduction in salt intake independent of baseline BMI. Second, in the sensitivity analysis using reduction in salt intake, measured by 24h urinary sodium, as the outcome among random sample of children, the association estimates were obtained after adjusting for age, gender and baseline BMI. These association showed consistent findings with the primary analysis, suggesting the robustness of the findings. We have clarified the relevant descriptions on the validation analyses (Line 27 on Page 6 in the method section; Line 13 on Page 10 in the result section) and the sensitivity analyses (Line 6-7 on Page 9 and supplementary table S1).

3.Line 53: Is patient ? a typo, perhaps participant

Response: We thank the reviewer for this comment. It is per the journal style that we should include a patient and public involvement statements. We agree with the reviewer that “participant involvement” is more appropriate for the current study, and we would like to change it accordingly if it follows the style of BMJ open.

4.Table 1& 3: why is it not possible to separate out mother from grandparents i.e. the ‘others’; fathers was collected separately and information can then be provided about the importance or not of support from this family member. Is there a reason why mothers was not specifically examined? Can this be stated in the method. Particularly as this is also a discussion point at line 41, page 12.

Response: We are grateful to the reviewer for the very helpful comment. It is possible to separate the role of mother from grandparents and we have revised it accordingly. Now we present the data for mother, father, grandmother and grandfather separately throughout the manuscript, including table 1, table 3, main text (Line 18-23 on Page 10) and Supplementary table S1. Previously we grouped the findings together based on the analyses for each family member, as we found that the association remains consistently significant with the largest magnitude for the support from the father, while the association for the support from mother, or grandparents attenuated substantially in the sensitivity analysis. We agree with the reviewer that we should separate these roles to provide enough information for the readers to understand the relationship in a clearer way. Given that the discrepancy in the association estimates could also be due to differential statistical power, we have toned down the related discussion and interpreted the findings with extra caution (Line 12-13 on Page 12).

Reviewer #3 Walter Lehmacher Uni Köln, no comments

Reviewer #4 Gabriel Escarela Universidad Autónoma Metropolitana - Iztapalapa

1.The methods are correctly used. I recommend to give the references of the techniques used and not only the software employed.

Response: Thank you for your helpful comment. We have added the references for the techniques used as suggested (Line 9-10 on Page 9).

Reviewer #5 Yipu Shi, Public Health Agency of Canada, Canada

Please leave your comments for the authors below Comments to the statistical procedures:

Well designed and methodological vigour school based intervention study.

1. Mixed linear models were used to appropriately account for the clustering effect of children in the same class. It's a methodological strength however, it seems that the association of SRB with each aspect of the children's social networks was assessed individually, then the joint effect of these networks on SRB was assessed in a new model by choosing one most influencing variable from each network (page 8 line 46), can't this be done in one mixed model so the independent effect of all variables of the network are assessed simultaneously?

Response: We thank the reviewer for the compliments on our study and for the thoughtful comments on the analysis approach. We fully agree with the reviewer that it is statistically possible to include all variables of all network factors in a single model. We prefer not to do so mainly because of the scientific questions of interest. Specifically, we are primarily interested in understanding the role of network factors for each of the family, peer and teacher networks separately. That said, this network-specific investigation is our primary analysis. The model, in which a representative factor of each network was examined simultaneously, is a secondary analysis to additionally explore which network (i.e. family, peer or teacher network) could be potentially more important. In this secondary analysis, only one variable for each network domain was finally chosen because of the largely overlapping interpretations of some variables within the same network domain. For example, for peer network, both network density (proportion of friendship connections within the same class) and normalized degree centrality (standardized number of friends per child) reflect the interconnectedness of children and it would be reasonable to choose one representative factor for each aspect. Additionally, network density and normalized degree centrality is closely related to each other and it could lead to the issue of collinearity and misspecification if they are put in the same model.

To clarify this issue, we now make it clear in our statistical analysis section that the final joint model is an additional analysis to further explore which network (family, peer and teacher) would be potentially more important for influencing the salt-reduction behaviors of children (Line 23-36 on Page 8). We also performed the sensitivity analysis in which the representative factors were replaced. Similar findings were observed if size and density of peer networks were used.

2. It's not clear how the family support (partial vs. full) was defined in the joint mixed linear model, looks like it's defined differently in the individual family support model (Table 3)

Response: We thank the reviewer for raising this very helpful comment. In the final joint mixed linear model, to facilitate an easier interpretation, the four categories of family support were grouped into two levels, i.e. partial and full support. Full support was defined as all family members supporting salt reduction. Partial support was defined as at least one family member not supporting salt reduction. We have added a footnote in Table 3 to clarify this issue.

3. Are the children having their 24-hour urine collected before and after the trial used for the validation of the SRB score (n=135) the same as those used for the sensitivity analysis (n=135)? Suggest the reduction in 24-hour urine used in the mixed model in Supplementary table S1 be treated as positive so that the direction and strengths of the association are comparable with Table 3.

Response: Yes. The children having their 24-hour urine collected before and after the trial used for the validation of the SRB score (n=135) were the same as those used for the sensitivity analysis

(n=135). As suggested, we have changed the reduction in 24-hour urine used in the mixed model in Supplementary table S1 as positive to make the interpretation consistent and comparable with Table 3. We have also put a footnote under Supplementary table S1 to give an example of the interpretation. Thank you for this thoughtful comment.

4. The authors used advanced statistical model and social network analysis techniques in addition to the traditional method to carefully examine the role of peer influence in salt reduction, which is another technical strength and contribution to the methodologies in this field.

Response: Thank you.

VERSION 2 – REVIEW

REVIEWER	Tatsuo Shimosawa International University of Health and Welfare, Japan
REVIEW RETURNED	29-Apr-2019

GENERAL COMMENTS	The revision is adequately done.
----------------------------------

REVIEWER	Dr. Yipu Shi Public Health Agency of Canada
REVIEW RETURNED	08-May-2019

GENERAL COMMENTS	The authors have adequately addressed all my questions.
---